# CCN2 Aggravates the Immediate Oxidative Stress–DNA Damage Response following Renal Ischemia–Reperfusion Injury

**DOI:** 10.3390/antiox10122020

**Published:** 2021-12-20

**Authors:** Floris A. Valentijn, Sebastiaan N. Knoppert, Georgios Pissas, Raúl R. Rodrigues-Diez, Laura Marquez-Exposito, Roel Broekhuizen, Michal Mokry, Lennart A. Kester, Lucas L. Falke, Roel Goldschmeding, Marta Ruiz-Ortega, Theodoros Eleftheriadis, Tri Q. Nguyen

**Affiliations:** 1Department of Pathology, University Medical Center Utrecht, 3584 CX Utrecht, The Netherlands; S.N.Knoppert-3@umcutrecht.nl (S.N.K.); R.Broekhuizen@umcutrecht.nl (R.B.); L.A.Kester@prinsesmaximacentrum.nl (L.A.K.); lucasfalke@hotmail.com (L.L.F.); R.Goldschmeding@umcutrecht.nl (R.G.); T.Q.Nguyen@umcutrecht.nl (T.Q.N.); 2Department of Nephrology, Faculty of Medicine, University of Thessaly, 412 22 Larissa, Greece; gpissas@msn.com (G.P.); teleftheriadis@yahoo.com (T.E.); 3Molecular and Cellular Biology in Renal and Vascular Pathology. Fundación Instituto de Investigación Sanitaria—Fundación Jiménez Díaz, Universidad Autónoma de Madrid, 28040 Madrid, Spain; RRodriguez@fjd.es (R.R.R.-D.); laura.marqueze@quironsalud.es (L.M.-E.); MRuizO@fjd.es (M.R.-O.); 4Department of Pediatric Gastroenterology, Wilhelmina Children’s Hospital, 3584 EA Utrecht, The Netherlands; M.Mokry@umcutrecht.nl

**Keywords:** acute kidney injury, ischemia–reperfusion injury, oxidative stress response, DNA damage response, cell cycle arrest, cellular communication network factor 2

## Abstract

AKI, due to the fact of altered oxygen supply after kidney transplantation, is characterized by renal ischemia–reperfusion injury (IRI). Recent data suggest that AKI to CKD progression may be driven by cellular senescence evolving from prolonged DNA damage response (DDR) following oxidative stress. Cellular communication factor 2 (CCN2, formerly called CTGF) is a major contributor to CKD development and was found to aggravate DNA damage and the subsequent DDR–cellular senescence–fibrosis sequence following renal IRI. We therefore investigated the impact of CCN2 inhibition on oxidative stress and DDR in vivo and in vitro. Four hours after reperfusion, full transcriptome RNA sequencing of mouse IRI kidneys revealed CCN2-dependent enrichment of several signaling pathways, reflecting a different immediate stress response to IRI. Furthermore, decreased staining for γH2AX and p-p53 indicated reduced DNA damage and DDR in tubular epithelial cells of CCN2 knockout (KO) mice. Three days after IRI, DNA damage and DDR were still reduced in CCN2 KO, and this was associated with reduced oxidative stress, marked by lower lipid peroxidation, protein nitrosylation, and kidney expression levels of Nrf2 target genes (i.e., HMOX1 and NQO1). Finally, silencing of CCN2 alleviated DDR and lipid peroxidation induced by anoxia-reoxygenation injury in cultured PTECs. Together, our observations suggest that CCN2 inhibition might mitigate AKI by reducing oxidative stress-induced DNA damage and the subsequent DDR. Thus, targeting CCN2 might help to limit post-IRI AKI.

## 1. Introduction

Acute kidney injury (AKI) that requires renal replacement therapy is associated with over 50% mortality, and the severity of AKI in hospitalized patients correlates to mortality, length of hospital stay, and healthcare costs [1,2]. In addition, AKI predisposes to chronic kidney disease (CKD) development later in life, in part due to the excessive fibrosis associated with tubular injury [3]. Ischemia–reperfusion injury (IRI) is a leading cause of AKI, which occurs in 30–50% of patients receiving deceased donor kidneys and, in this context, often leads to delayed graft function [4,5]. Understanding the immediate pathophysiological changes that underlie IRI in the acute phase is paramount for identifying early intervention options to limit subsequent AKI to CKD progression.

During renal IRI, excessive amounts of reactive oxygen species (ROS) are produced [6,7]. ROS induce DNA damage and, consequently, lead to a DNA damage response (DDR) to facilitate cell repair and regeneration of the injured tubules through DNA repair or apoptosis [8,9]. However, a subset of cells fails to go into apoptosis, despite unresolved DNA damage, resulting in persistent DDR and senescence [10]. Senescent cells alter their microenvironment through a proinflammatory and profibrotic secretome dubbed the senescence-associated secretory phenotype (SASP), thereby affecting organ morphology and function [11]. Preventing cellular senescence by interfering with senescent-associated intracellular pathways has been proposed as a novel strategy to prevent progression to CKD [12].

Cellular communication network factor 2 (CCN2), previously known as connective tissue growth factor (CTGF), is a well-known modulator in CKD development [13,14]. CCN2 is a matricellular protein involved in various processes including matrix biology, angiogenesis, inflammation, cell proliferation, hypoxia sensing, senescence, and apoptosis [15,16,17,18,19]. Furthermore, CCN2 is associated with DDR and subsequent cellular senescence in delayed graft function and chronic allograft dysfunction following kidney transplantation and in experimental IRI [20], and it is also a potent inducer of vascular oxidative stress [21].

Differential regulation of gene expression in multiple signaling pathways associated with CCN2 in the immediate response, 4 h after experimental IRI led us to hypothesize that CCN2 can contribute to oxidative stress-induced DDR in the immediate phase during IRI. To elucidate this hypothesis, we assessed kidney function, damage, DDR, and oxidative stress markers in wild-type (WT) and CCN2 knockout (KO) mice 4 h and 3 days after IRI. In addition, we also evaluated the effect of CCN2 treatment in mice and of CCN2 silencing in an IRI-like model in cultured renal epithelial cells.

## 2. Materials and Methods

### 2.1. Animals

Experiments were performed according to the European community guidelines for animal experiments and the ARRIVE guidelines and with consent of the Experimental Animal Ethics Committee of the University of Utrecht, The Netherlands [22]. Generation of tamoxifen-inducible CCN2 full-knockout mice is extensively described elsewhere [23]. In brief, CCN2^Flox/Flox^ mice were crossbred with ROSA26CreERT2 mice (Gt(ROSA)26Sor^tm(cre/ERT2)Tyj^/J, The Jackson Laboratory, Maine, USA), both on a C57Bl6/J background. Male 12–14 week old mice received four intraperitoneal injections on alternate days over a course of 7 days with either 100 μL corn oil (Sigma–Aldrich, St. Louis, MO, USA) or 100 μL of 10 mg/mL tamoxifen–corn oil solution (Sigma–Aldrich, St. Louis, MO, USA). After the last injection, a 14 day washout period was followed by the IRI operation. Sample sizes were based on published studies and pilot experiments. [24] 

### 2.2. Ischemia–Reperfusion Injury Model

The surgical procedures were executed as previously described [24]. In short, renal pedicles were located through an abdominal midline incision and bilaterally clamped for 25 min with atraumatic neurovascular clamps. Clamping and subsequent reperfusion associated color changes were visually confirmed. IRI mice without color changes were excluded from further analyses. Sham operated animals underwent the same procedure with the exception of the pedicle clamping. The operations were executed under 2% isoflurane anesthesia, and body temperature was maintained at 37 °C. After 4 h, mice were euthanized by a lethal dose of a ketamine–xylazine–acepromazine injection, and the plasma and organs were collected and stored at −80 °C.

### 2.3. Mouse Recombinant CCN2 Model

CCN2 administration studies were performed in 3 month old male C57Bl/6 mice as previously described [25]. Mice received 2.5 ng/g body weight recombinant human CCN2 (corresponding to the C-terminal module IV of CCN2; MBL International Corporation, Woburn, MA, USA) or saline by intraperitoneal administration.

### 2.4. Cell Culture

Mouse primary renal PTECs (RPTECs, cat. no. C57-6015, Cell Biologics, Chicago, IL, USA) were cultured as described previously using a GasPak EZ Anaerobe Container System with Indicator (cat. no. 26001, BD Biosciences, S. Plainfield, NJ, USA) [26]. Gene silencing was performed using a predesigned siRNA corresponding to CCN2 (siRNA ID MSS274358, Thermo-Fisher, Waltham, MA, USA) according to the manufacturer’s protocol. In short, sub-confluent cells were transfected in Opti-MEM reduced serum medium (Invitrogen, Waltham, MA, USA) for 24 h with 5 ng/mL siRNA using 50 nM lipofectamine RNAi-MAX (Invitrogen, Waltham, MA, USA), followed by a 24 h incubation in 20% FBS medium and a 24 h incubation in serum-free medium. Controls were non-transfected cells treated with lipofectamine vehicle. Cells were subjected to 24 h of anoxia and 2 h of reoxygenation.

### 2.5. Plasma Urea

Plasma urea was measured using the colorimetric assay conform manufacturer’s protocol (DiaSys, Holzheim, Germany).

### 2.6. Histology and Immunohistochemistry

Renal tissue was fixed in a buffered 4% formalin solution for 24 h and, subsequently, embedded in paraffin blocks. Sections were mounted on adhesive slides (Leica Xtra) and rehydrated through a series of xylene and alcohol washes after which slides were rinsed in distilled water.

For periodic acid-Schiff (PAS) staining, standard procedures were used (Dako, Glostrup, Denmark). Immunohistochemistry (IHC) for gamma H2AX (γH2AX), 4-hydroxynonenal (4-HNE), and Bcl-xL were performed as described elsewhere [27,28,29]. IHC for serine 15 phosphorylated p53 (p-p53) and nitrotyrosine was set-up based on the manufacturer’s protocol. First, endogenous peroxidase was blocked using H_2_O_2_, followed by antigen retrieval by boiling in pH6 citrate buffer and primary antibody incubation for two hours at room temperature (anti-γH2AX (p Ser139), Novus Biologicals NB100-2280, 1:500; anti-p-p53 (p Ser15), R&D Systems AF1043, 1:800; anti-Bcl-xL, Abcam ab178844, 1:16,000; anti-nitrotyrosine, Bioss bs-8551R, 1:500) or overnight at 4 °C (anti-4-HNE, Abcam ab46545, 1:150), diluted in 1% BSA blocking solution. For 4-HNE IHC, endogenous peroxidase was blocked after the primary antibody to prevent extra lipid oxidation. For p-p53, Bcl-xL, 4-HNE, and nitrotyrosine, secondary HRP-conjugated antibodies were applied and visualized using a Nova Red substrate (Vector Laboratories, Burlingame, CA, USA). For γH2AX, alkaline-phosphatase-conjugated antibody and liquid permanent red substrate (Dako) were used. Slides were counterstained with Mayer’s hematoxylin.

Slides were scanned (NanoZoomer, Hamamatsu, Japan) and images were acquired by taking photographs in ImageScope. For the assessment of ATN severity, a blinded pathologist (TQN)-graded cortical ATN severity using an arbitrary score ranging from 0 (i.e., none) to 3 (i.e., severe) on PAS-stained slides. ATN was scored based on loss of TECs, luminal debris, and mitotic activity. The score was displayed as the mean of the left and right kidney. Nuclear expression of γH2AX and p-p53 was quantitated in QuPath 0.2.3 (Edinburgh, Scotland) by counting the number of IHC-positive cells, relative to the number of hematoxylin stained nuclei in whole slides [30]. Total expression of p-p53, 4-HNE, nitrotyrosine, and Bcl-xL was quantitated in QuPath by setting a pixel classifier based on selected annotations corresponding to positive and negative areas, resulting in the percentage of positive area relative to total tissue area in whole slides.

### 2.7. RNA-Seq

Full transcriptome RNA-sequencing was performed on a cDNA library constructed from RNA extracted from 18 selected samples of kidney cortex (4 WT sham; 4 WT IRI; 4 KO sham; 6 KO IRI). mRNA was isolated using NEXTflex Poly(A) Beads (Bioo Scientific, Austin, TX, USA). Libraries were prepared using the Rapid Directional RNA-Seq Kit (NEXTflex), looped in equimolarly and sequenced on Illumina NextSeq500 to produce single-end 75 base long reads, and read-count analysis was performed by the Utrecht DNA Sequencing Facility (Utrecht, The Netherlands). RNA-seq reads were aligned to the reference genome GRCm38 using STAR version2.4.2a (Cold Spring Harbor, NY, USA) [31]. Read groups were added to the BAM files with Picard’s AddOrReplaceReadGroups v1.98 (Broad Institute, Cambridge, MA, USA). The BAM files were sorted with Sambamba v0.4.5, and transcript abundances were quantified with HTSeq-count version0.6.1p1 using the union mode [32]. Subsequently, reads per kilobase of transcript per million mapped reads (RPKM’s) were calculated with edgeR’s RPKM function [33]. The full data set with RPKM values was deposited in the GEO database (accession number: GSE186316).

Differentially expressed genes were identified using the DESeq2 package with standard settings [34]. False discovery rate (FDR) adjusted *p*-values were used to determine significance [35]. Pathway enrichment analysis was performed using Gene Set Enrichment Analysis (GSEA; MSigDB, UC San Diego, CA, USA) [36]. GSEA analysis was performed on the expression data set file containing quantile normalized log2 transformed RPKM values, the Hallmarks and KEGG gene sets databases and the “Mouse_Symbol_with_Remapping_to_Human_Orthologs” chip platform. Gene sets with an FDR < 25% were identified as significantly enriched and ranked based on the normalized enrichment score (NES). A full list of differentially expressed genes (DEGs) and enriched pathways between all groups is included in Appendix A.

### 2.8. Quantitative Real-Time PCR

Full RNA was extracted from kidney cortical poles using TRIzol Reagent (Thermo-Fisher, Waltham, MA, USA). RNA purity and quantity were determined using the NanoDrop 2000 (Thermo-Fisher). Using 3 μg RNA per kidney, a cDNA library was synthesized with SuperScript III reverse transcriptase (Thermo-Fisher). Relative RNA expression levels were determined on a ViiA 7 real-time PCR system (Applied Biosystems, Pleasanton, CA, USA). The SYBR green primer sequences and TaQman probes used for quantitative real-time polymerase chain reaction (qPCR) are shown in Appendix A. TATA-box binding protein (TBP) was used as internal reference gene. Samples were run in duplicate. Samples free of mRNA and reverse transcriptase were used to control for potential contamination of the reaction. The ΔΔCT method was used to calculate relative expression levels.

### 2.9. Western Blot

Cultured RPTECs were lysed and Western blot analysis was routinely performed as described previously [26]. Experiments were repeated three times. Membranes were incubated with antibodies specific for the following proteins: 4-HNE (1:500; Abcam ab46545), nitrotyrosin (1:500; Bioss bs-8551R; Bcl-xL (1:500; Abcam ab178844), HMGB1 (1:500; Proteintech 10829-AP-1), p-p53 (1:1000, Cell Signaling Technology #9284), and β-actin (1:2500; Cell Signaling Technology #4967).

### 2.10. MDA Assay

Lipid peroxidation was assessed in RPTECs cultured in 6-well plates. At the end of the reoxygenation period, malondialdehyde (MDA), the end product of lipid peroxidation, was measured fluorometrically in cell extracts with the Lipid Peroxidation (MDA) Assay Kit (cat. no. ab118970, Abcam, Cambridge, UK). Before MDA measurement, a Bradford assay was performed to adjust lysate volumes to equal protein concentration. These experiments were performed six times.

### 2.11. Statistical Analysis

Two-way ANOVA with post hoc Tukey correction was used to compare means of the various parameters in all 4 groups at both time points. Discrete dependent variables were tested non-parametrically with the Kruskal–Wallis test (ATN grade). Correlation of two independent variables was assessed using Pearson. Values exceeding >1.5 interquartile ranges from the mean were labeled as outliers and excluded. Data that showed abnormal distribution (i.e., right skewness) were log-transformed. Otherwise, normal distribution was assumed. The homogeneity of variances was tested with Levene’s test because of unequal sample sizes. All statistical analyses were executed using the statistical program SPSS (IBM SPSS Statistics 25). Error bars represent SEM, and *p*-values smaller than 0.05 were considered statistically significant.

## 3. Results

### 3.1. CCN2 Deletion Did Not Reduce IRI-Induced Functional Decline and Acute Tubular Necrosis 4 h after IRI

To evaluate CCN2-mediated changes in immediate IRI, a 4 h IRI model was conducted using tamoxifen-inducible CCN2 full KO mice (Figure 1A). Tamoxifen-induced recombination resulted in 77% reduction in CCN2 mRNA in both sham and IRI kidneys (*p* = 0.005 and *p* = 0.008, respectively; Appendix A). Plasma urea increase upon IRI was similar in WT and in CCN2 KO mice (Appendix A). Histological examination revealed IRI-induced acute tubular necrosis, but no significant difference between WT- and CCN2 KO IRI mice (Appendix A). Concordantly, tubular injury markers KIM-1, NGAL, and SOX9 were equally upregulated in CCN2 KO IRI and WT IRI kidneys (Appendix A).

### 3.2. CCN2 Regulated Gene Transcription of Several Signaling Pathways 4 h after IRI

To find CCN2-related changes in primary major regulatory events associated with the immediate response to IRI in the renal cortex, full transcriptome RNA-sequencing was performed. According to the literature, distinct differential gene expression patterns can be recognized in the kidney, as early as three hours after IRI, but the increase of a pro inflammatory transcriptome in those reports was limited [37,38]. In order to allow sufficient time for alteration of transcriptome profile with limited risk of analyzing secondary regulatory mechanisms, we harvested kidneys 4 h after reperfusion.

Principal component analysis (PCA), as a method to identify similarities and differences in overall gene expression, showed that the data clearly segregated WT IRI, CCN2 KO IRI, and SHAM groups, but no evident sub-clusters formed when comparing WT SHAM with CCN2 KO SHAM mice (Figure 1B). Pairwise comparison of transcriptome reads between WT IRI and CCN2 KO IRI mice yielded a total of 4167 differentially expressed genes (DEGs) (Appendix A, of which 2169 genes were significantly downregulated and 1998 upregulated in CCN2 KO IRI mice (Figure 1C). DEGs with the largest fold down- or upregulation are presented in Appendix A. In sham mice, a total of 21 DEGs were identified of which 15 downregulated and six upregulated in CCN2 KO SHAM compared to WT SHAM (Figure 1C).

Gene set enrichment analysis (GSEA) on DEGs in CCN2 KO IRI compared to WT IRI mice using the Hallmarks and the Kyoto Encyclopedia of Genes and Genomes (KEGG) databases identified significant enrichment of 16 pathways with the WT IRI phenotype (Appendix A) and 18 pathways with the KO IRI phenotype (Appendix A) [39]. WT IRI phenotype-enriched pathways included cytokine (TNFα and IL-6/JAK/STAT3), growth factor (TGFβ), and downstream transcription factor signaling (p53, PI3K/AKT/mTOR and mTORC1) as well as inflammatory response, hypoxia, and oxidative phosphorylation, suggesting overall lower activation of these pathways in CCN2 KO compared to WT IRI-mice.

### 3.3. CCN2 Deletion Reduced DNA Damage and p53 Activation 4 h after IRI

Oxidative stress in IRI involves excessive ROS production, driving DNA damage and a DDR involving p53 pathway’s activation [8,40,41]. Markers related to DNA damage, p53 signaling, and oxidative stress were analyzed by immunohistochemistry (IHC) and qPCR. Lower cell numbers positive for the double-strand break marker phosphorylated H2AX (γH2AX) were observed in CCN2 KO IRI kidneys compared to WT IRI kidneys, indicating reduced DNA damage (*p* < 0.005; Figure 2A). Additionally, p-p53 expression, indicative for DNA damage-induced p53 activation [42], was reduced by CCN2 KO (*p* = 0.01; Figure 2B). Furthermore, in CCN2 KO IRI kidneys, CCN2 mRNA correlated with p-p53 (*r* = 0.77; *p* = 0.04; Figure 2B). Of note, γH2AX and p-p53 were mainly expressed in tubular epithelial cells (TECs). However, although we observed upregulation of p53 transcriptional target genes GADD45A, an established immediate IRI response gene [43,44] and p21Cip1 (p21), CCN2 KO did not reduce the expression level of these genes on this time point (Figure 2C,D).

### 3.4. CCN2 Deletion Reduced DDR and Apoptosis Resistance 3 Days after IRI

To further evaluate the protective effects of CCN2 deletion, we performed a 3 day IRI model. In this model, we previously observed less DNA damage and p21 upregulation in CCN2 KO kidneys compared to WT kidneys 3 days after IRI [20]. Furthermore, CCN2 KO IRI kidneys had lower numbers of p-p53 positive cells (*p* = 0.02; Figure 3A) and reduced expression of GADD45A (*p* < 0.005; Figure 3B) 3 days after IRI. Expression of p-p53 was predominantly nuclear and mainly observed in TECs (Figure 3A,B). Additionally, gene and protein expression levels of anti-apoptotic mediator Bcl-xL (*p* = 0.04 and *p* = 0.02; respectively; Figure 4A,B) and gene expression level of HMGB1 (*p* < 0.005; Figure 4C) were reduced in CCN2 KO injured mice.

### 3.5. CCN2 Deletion Reduced Oxidative Stress Response 3 Days after IRI

The role of CCN2 in oxidative stress response was evaluated by assaying ROS production and redox signaling. The lipid peroxidation product 4-hydroxynonenal (4-HNE) and protein nitrosylation product nitrotyrosine, that mark ROS tissue levels, were not expressed 4 h after IRI (data not shown). IRI induced upregulation of ROS scavenging erythroid 2-related factor 2 (Nrf2 target) genes heme oxygenase 1 (HMOX1) and glutamate–cysteine ligase catalytic subunit (GCLC) (*p* = 0.009 and *p* < 0.005, respectively; Appendix A) and tendencies for increased expression after IRI were observed for (NAD(P)H quinone dehydrogenase 1 (NQO1) and glutamate–cysteine ligase modifier subunit (GCLM), but no differences in expression levels were observed between CCN2 KO and WT IRI kidneys (Appendix A–D).

However, in the later model, 3 days after IRI, CCN2 KO kidneys showed lower 4-HNE expression (Figure 5A; *p* < 0.005) and lower nitrotyrosine expression (Figure 5B; *p* < 0.005) indicating that CCN2 KO reduced excessive ROS production upon IRI. Expression of these oxidative stress markers was mainly observed in proximal tubules of the inner cortex. Furthermore, reduced mRNA levels of Nrf2 target genes HMOX1 and NQO1 were observed in CCN2 KO compared to WT IRI kidneys (*p* < 0.001 and *p* = 0.001; Figure 5C,D).

### 3.6. CCN2 Induced Oxidative Stress in Mouse Kidneys

To evaluate if CCN2 itself can induce an oxidative stress response, mice were injected with recombinant CCN2 and kidneys were harvested 24 h later. Kidneys from CCN2 injected mice showed an increase in 4HNE expression compared to control mice. (*p <* 0.05; Figure 6). Expression of 4-HNE was mainly observed in tubules (Figure 6).

### 3.7. CCN2 Silencing Reduces Oxidative Stress Induced by Anoxia–Reoxygenation Injury in PTECs

The direct impact of CCN2 on oxidative stress in renal tubular cells was evaluated by silencing CCN2 in primary murine RPTECs subjected to anoxia–reoxygenation (AR) injury, mimicking in vivo IRI [45], and showing a similar immediate upregulation of CCN2 [20]. AR-injured RPTEC showed more expression of p-p53, 4-HNE, and nitrotyrosine compared to control cells that were cultured under normoxic conditions, and p-p53 and 4-HNE expression were alleviated when CCN2 was silenced (Figure 7A). No marked differences in Bcl-xL or HMGB1 expression were observed (data not shown). Finally, AR-injured RPTEC showed increased levels of lipid peroxidation marker MDA compared to controls cells that were cultured under normoxic conditions, and this was alleviated when CCN2 was silenced (*p* < 0.005; Figure 7B). Taken together, silencing of CCN2 alleviated DDR and lipid peroxidation induced by anoxia–reoxygenation in-jury in cultured PTECs.

## 4. Discussion

Our observations in the immediate early response phase following experimental IRI identify CCN2-dependent enrichment of several signaling pathways involved in the stress response to IRI. This was evidenced by differential gene expression at 4 h after reperfusion, and follow-up analyses revealed that CCN2 deletion reduces DNA damage and DDR already at that time point, despite similar decline in kidney function and acute tubular damage. Subsequently, reduced DDR, anti-apoptosis and oxidative stress response were observed 3 days after IRI. Moreover, CCN2 silencing also reduced DDR and oxidative stress in cultured PTECs and intraperitoneally injected recombinant CCN2 induced oxidative stress in murine kidneys. CCN2 has previously been shown to contribute to vascular ROS generation [21], and to DDR and subsequent cellular senescence in delayed graft function and chronic allograft dysfunction following kidney transplantation, and in experimental IRI [20]. These novel findings suggest that CCN2 not only acts as a profibrotic factor in later stages of adverse tissue remodeling, but that it also negatively contributes to the immediate early response to IRI, in particular to oxidative stress-induced DDR (Figure 8). This could explain our previous findings that CCN2 deletion preserves kidney function 3 days after IRI, and contribute to reduced cellular senescence and fibrosis at later stages [20]. This implicates that anti-CCN2 therapy, currently in phase 2 and phase 3 trials for other indications, may be developed to reduce post-IRI acute and chronic kidney dysfunction by limiting IRI-induced oxidative stress induced DNA damage, senescent cell accumulation, and subsequent fibrosis.

This study provides novel insights into the effect of CCN2 on the oxidative stress response following renal IRI. Few studies have previously observed that CCN2 is a potent inducer of oxidative stress in other organs and cell types. CCN2 induces oxidative stress in murine aorta and in cultured vascular smooth muscle cells and endothelial cells [21]. Furthermore, CCN2 induces accumulation of ROS in cultured fibroblasts [46]. However, to our knowledge, this is the first report regarding CCN2-induced oxidative stress in the kidney, and the first demonstration of a protective effect of CCN2 inhibition on oxidative stress induced by other factors. Remarkably, CCN2 overexpression has also been described to limit oxidative stress. For instance, CCN2 protected cultured cardiac myocytes from doxorubicin-induced oxidative stress and cell death [47]. Possibly, the complex and diverse mechanisms of action of CCN2, might lead to different outcomes depending on the cell type and context, including matrix, cytokine environment, and genotype [48].

Main mechanisms by which CCN2 might contribute to oxidative stress following renal IRI include increased mitochondrial oxidative phosphorylation and ROS generation, and decreased ROS scavenging. First, CCN2 may interfere with mitochondrial oxidative phosphorylation, which is implicated in renal IRI [49]. GSEA revealed enrichment of the oxidative phosphorylation pathway 4 h after IRI in the direction of lower activation in CCN2 KO IRI- compared to WT IRI-mice. Consistently, in murine chondrocytes, CCN2 deletion impaired metabolism with reduced intracellular ATP levels [50]. Upon reperfusion of the kidney, restored levels of oxygen stimulate mitochondrial oxidative phosphorylation to produce ATP, the main source of cellular energy, with the concurrence of harmful ROS [6,51]. Thus, altered ATP production depending on CCN2 expression level could affect the level of ROS and subsequent DNA damage upon renal IRI. However, a contrary effect of CCN2 on ROS generation has also been described in Oral Squamous Cell Carcinoma cells, where CCN2 overexpression impaired mitochondrial oxidative phosphorylation [52]. A possible effect of CCN2 inhibition on oxidative phosphorylation and underlying mechanisms in the setting of renal IRI remains to be elucidated.

Underlying pathways linking CCN2 and the oxidative stress response may include epidermal growth factor receptor (EGFR) activation and NAD(P)H oxidases (Nox)1 activity, which have previously been linked to CCN2-induced ROS generation [21]. Crosstalk between CCN2 and ROS producing metabolic pathways may also involve integrin interactions. CCN1 induces ROS through integrin signaling [53], and engagement of integrin α6β1 and cell surface heparan sulfate proteoglycans (HSPGs) [46]. The mechanisms of action of CCN2 and CCN1 may well be similar in several respects, since they are highly homologous and bind the same receptors [53,54].

Alternatively, reduced CCN2 might improve the handling of ROS by antioxidant mechanisms. However, regarding Nrf2 target genes, no differences in gene expression levels of GCLC and GCLM were observed, and reduced rather than increased expression levels of HMOX1 and NQO1 were found in the CCN2-depleted mice.

## 5. Conclusions

In aggregate, our observations indicate that DNA damage and DDR in the kidney following IRI are mitigated by CCN2 inhibition through reduction of the oxidative stress response in tubular epithelial cells. Anti-CCN2 therapy might therefore be explored for its potential to help limiting post-IRI AKI and AKI to CKD progression.

## Figures and Tables

**Figure 1 antioxidants-10-02020-f001:**
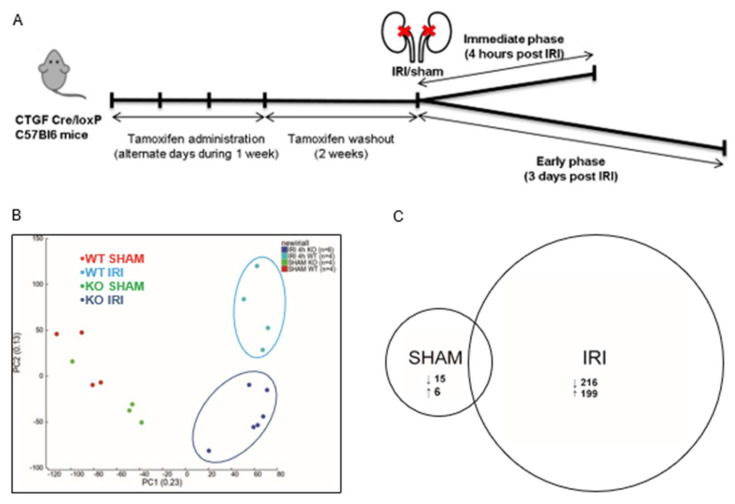
CCN2 regulated gene transcription 4 h after renal IRI. (**A**) Time course of the IRI mouse model; (**B**) principal component analysis plot of sham and IRI RNA sequence data in both WT and CCN2 KO genotypes showed clustering of WT IRI, CCN2 KO IRI, and sham groups but no clustering within the sham group; (**C**) pairwise comparisons of RNA sequence data of sham and IRI data in both WT and CCN2 KO genotypes showing the amount of differentially expressed genes.

**Figure 2 antioxidants-10-02020-f002:**
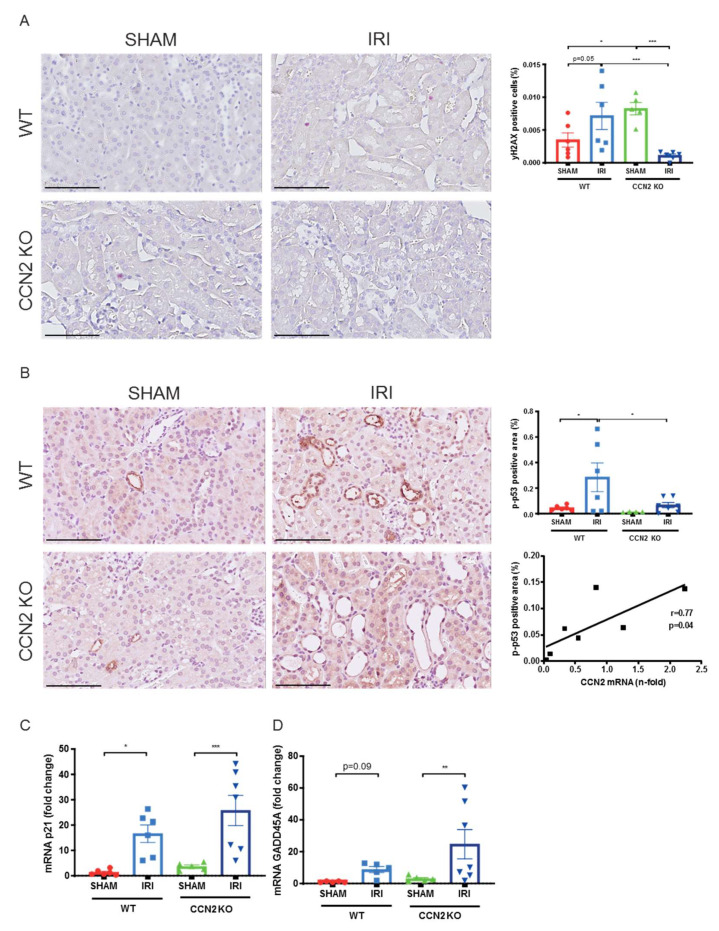
Near total deletion of CCN2 resulted in reduced DNA damage and p53 activation 4 h after IRI. (**A**,**B**) Representative micrographs of mouse renal cortex stained with gamma H2AX (γH2AX; (**A**) and phosphorylated p53 (p-p53; **B**). Quantification of γH2AX (**A**) and p-p53 (**B**) staining showed that increased DNA damage and p53 activation in IRI kidneys decreased in CCN2 KO mice compared with WT mice. Additionally, p-p53 expression correlated with CCN2 mRNA in CCN2 KO kidneys (**B**). (**C**,**D**) Quantitative real-time polymerase chain reaction (qPCR) analysis showed that mRNA expression levels of p21CIP1 (p21) and GADD45A increased by IRI in WT and in CCN2 KO kidneys. Data are expressed as the mean ± SEM (*n*= 6 for WT sham; *n* = 6 for WT IRI; *n* = 4–5 for KO sham; *n* = 6–7 for KO IRI). TBP was used as an internal control. * *p* < 0.05, ** *p* < 0.01, and, *** *p* < 0.005. Bar = 100 µm.

**Figure 3 antioxidants-10-02020-f003:**
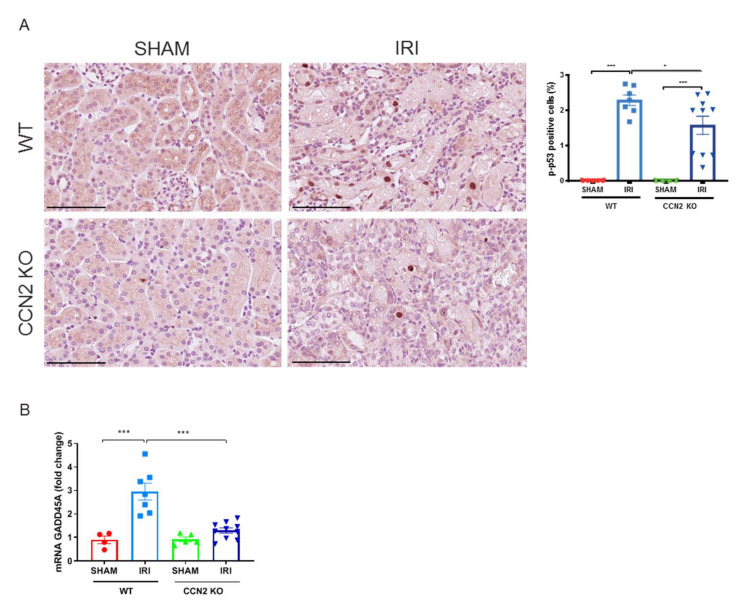
Near total deletion of CCN2 resulted in reduced p53 activation 3 days after IRI. (**A**) Representative micrographs of mouse renal cortex stained with phosphorylated p53 (p-p53). Quantification of p-p53 staining showed that increased p53 activation in IRI kidneys, decreased in CCN2 KO mice compared with WT mice. (**B**) qPCR analysis showed that increased mRNA expression of GADD45A in IRI kidneys reduced in CCN2 KO mice compared with WT mice. Data are expressed as the mean ± SEM (*n* = 4–5 for WT sham; *n* = 7 for WT IRI; *n* = 4–5 for KO sham; *n* = 10 for KO IRI). TBP was used as an internal control. * *p* < 0.05 and *** *p* < 0.001. Bar = 100 µm.

**Figure 4 antioxidants-10-02020-f004:**
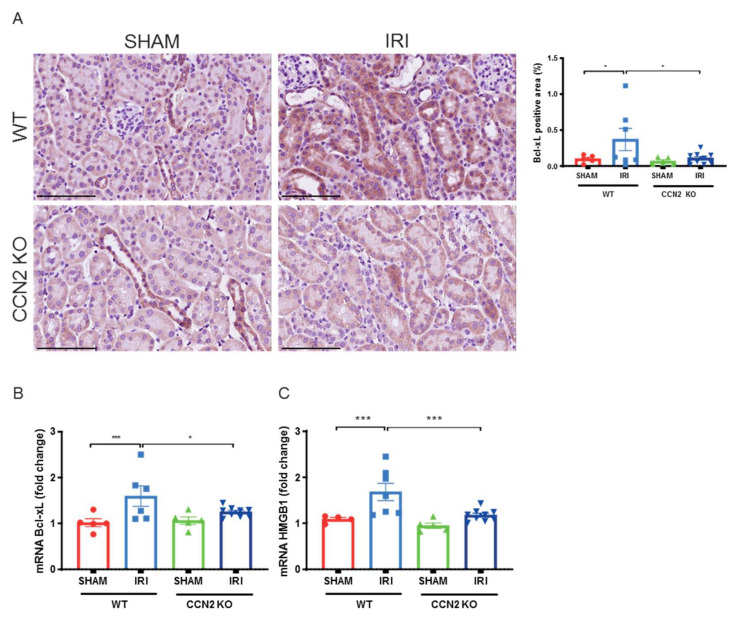
Near total deletion of CCN2 expression resulted in reduced Bcl-xL and HMGB1 anti-apoptosis expression 3 days after IRI. (**A**) Representative micrographs of mouse renal cortex stained with Bcl-xL showed that increased Bcl-xL expression in IRI kidneys decreased in CCN2 KO mice compared with WT mice. (**B**,**C**) qPCR analysis showed that increased mRNA expression of Bcl-xL (**B**) and HMGB1 (**C**) in IRI kidneys reduced in CCN2 KO mice compared with WT mice. Data are expressed as the mean ± SEM (*n* = 4–5 for WT sham; *n* = 7 for WT IRI; *n* = 5 for KO sham; *n* = 9–10 for KO IRI). TBP was used as an internal control. * *p* < 0.05 and *** *p* < 0.001. Bar = 100 µm.

**Figure 5 antioxidants-10-02020-f005:**
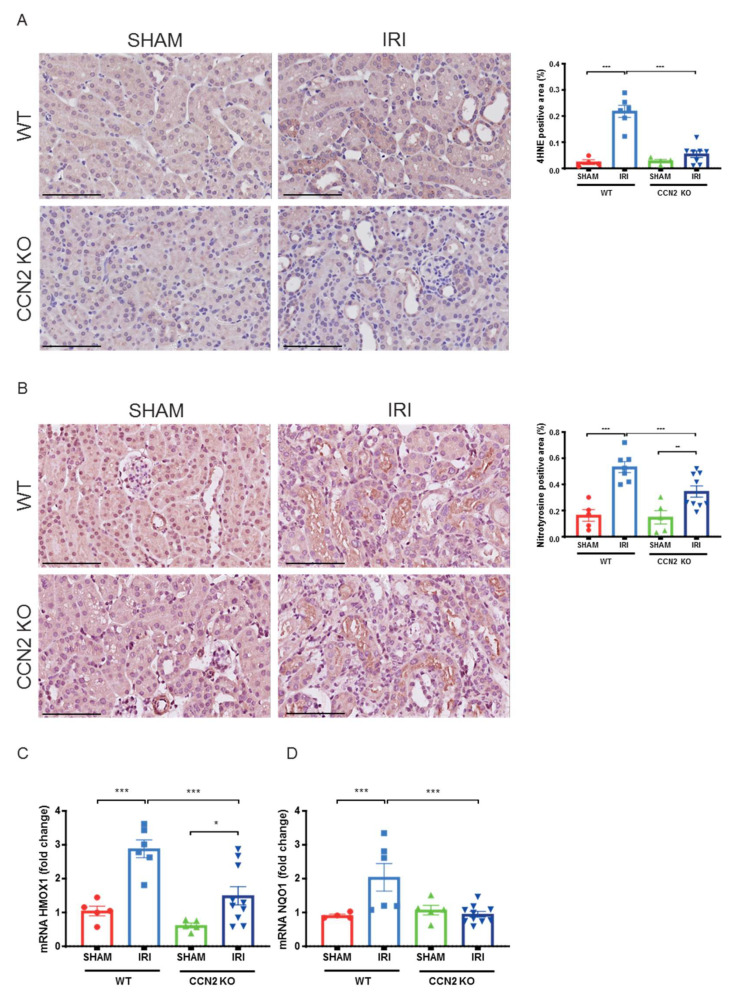
Near total deletion of CCN2 expression resulted in reduced oxidative stress response 3 days after IRI. (**A**,**B**) Representative micrographs of mouse renal cortex stained with 4-hydroxynonenal (4-HNE; **A**) and nitrotyrosine (**B**). Quantification of 4-HNE (**A**) and nitrotyrosine (**B**) staining showed that an IRI-induced increase of lipid peroxidation and protein nitrosylation, respectively, decreased in CCN2 KO mice compared with WT mice. (**C**,**D**) qPCR analysis showed that increased mRNA expression of HMOX-1 (**C**) and NQO1 (**D**) in IRI kidneys significantly reduced in CCN2 KO mice compared with WT mice. Data are expressed as the mean ± SEM (*n* = 4–5 for WT sham; *n* = 6–7 for WT IRI; *n* = 4–5 for KO sham; *n* = 8–10 for KO IRI). TBP was used as an internal control. * *p* < 0.05, ** *p* < 0.01, and *** *p* < 0.001. Bar = 100 µm.

**Figure 6 antioxidants-10-02020-f006:**
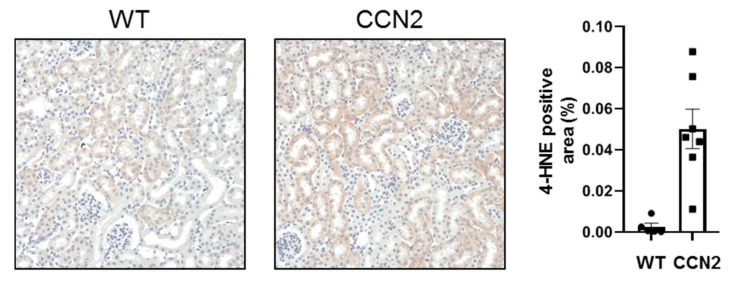
**Administration of CCN2 induced oxidative stress in murine kidneys.** Mice received intraperitoneal injections with 2.5 ng/g of CCN2 and were sacrificed 24 h later. Representative micrographs and quantification of mouse renal cortex stained with 4-hydroxynonenal (4-HNE), showing 4-HNE positive tubules. Data are expressed as the mean ± SEM of 4-HNE positive area per field (*n* = 5 for control and *n* = 7 for CCN2).

**Figure 7 antioxidants-10-02020-f007:**
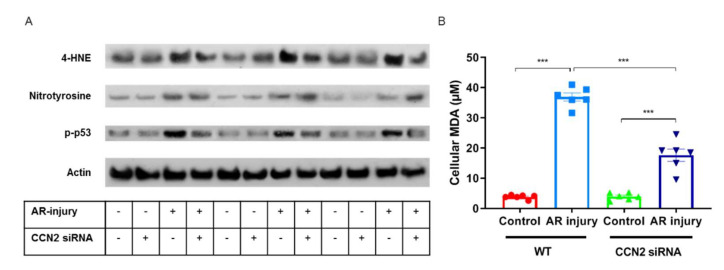
Silencing of CCN2 alleviated oxidative stress response induced by AR-injury in cultured PTECs. (**A**) Representative Western blotting of protein extracts from cultured cells for 4-HNE, nitrotyrosine, and p-p53 showed that increased oxidative stress response and DDR in AR-injured PTECs was alleviated by silencing of CCN2. Data represent three independent experiments. (**B**) Malondialdehyde (MDA) levels indicated that increased oxidative stress response in AR-injured PTECs was alleviated by silencing of CCN2. Data represent six independent experiments. Data are expressed as the mean ± SEM. *** *p* < 0.001.

**Figure 8 antioxidants-10-02020-f008:**
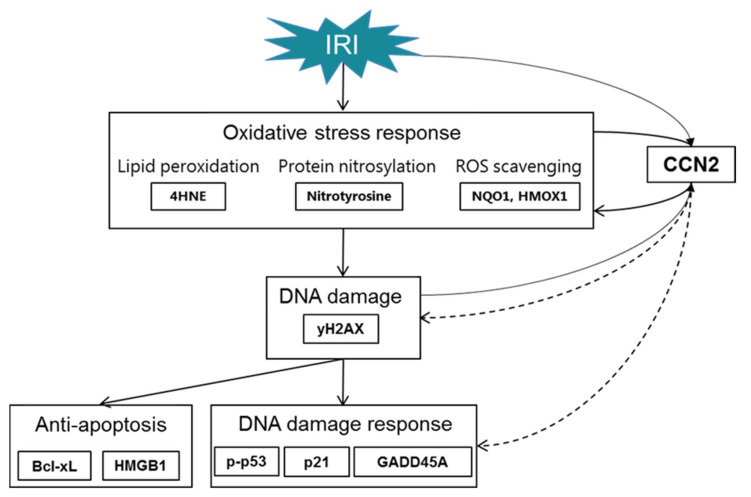
Mechanisms involved in IRI linked with CCN2. Inhibition of CCN2 reduced the expression of γH2AX and p-p53 marking reduced DNA damage and DDR in the immediate early response following IRI. In the early phase upon IRI, this was followed by sustained reduction in DNA damage and DDR, along with reduced anti-apoptosis. This was associated with reduced oxidative stress response. Dotted arrows are used to acknowledge that the exact mechanisms by which CCN2 contributes to DNA damage and DDR remain to be established.

## Data Availability

The RNA-seq data presented in this study are openly available in the GEO database (accession number: GSE186316).

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
