# Peer review of "CCN2 Aggravates the Immediate Oxidative Stress–DNA Damage Response following Renal Ischemia–Reperfusion Injury"

_antioxidants, 2021, doi:10.3390/antiox10122020_

Round 1

Reviewer 1 Report

In the manuscript, the authors investigated the effect of cellular communication factor 2 (CCN2) on oxidative stress using a mouse model of ischemia-reperfusion-induced AKI as well as a cell model of anoxia-reoxygenation-injured renal epithelial cells. The results showed that inhibition of CCN2 attenuated oxidative stress induced DNA damage and DNA damage response. The findings provide some useful information about the role of CCN2 in ischemia-reperfusion-induced AKI. However, there are some major issues as listed below:

  1. As shown in the manuscript, CCN2 deletion does not reverse IRI-induced functional decline and tubular injury 4 hours after IRI. However, it’s not clear if CCN2 deletion improved renal function and tubular injury at 3 days after IRI. It would be more convincing to include the relevant information in the manuscript if not published previously.
  2. The authors claimed that some important findings such as CCN2 deletion preserves kidney function and reduces fibrosis at 3 days after IRI were published in their previous study (Reference 20). However, reference 20 is a manuscript that was submitted but not published yet, which can not be used as a reference.
  3. It’s not clear why different lipid peroxidation marker (MDA for in vitro and 4-Hydroxynonenal for in vivo) were used. It would be more convincing to include the markers for oxidative stress (such as HMOX1, NQO1), anti-apoptosis (such as Bcl-xL and HMGB1) and DNA damage response (P53) from the cell model as did in the in vivo model to further confirm the in vivo findings.
  4. For Figure 2A the immunohistochemistry staining of γH2AX, it seems that there are not any γH2AX positive cells in all the groups except WT IRI group. Better representative images need to be included.
  5. Figure 2 C and D are missing.
  6. The full names for Nrf2, HMOX1, NQO1, HMGB1, GCLC, GCLM are missing.

Reviewer 2 Report

Valentijn et al. studied the role of CCN2 protein in DNA damage and subsequent cellular senescence and fibrosis after renal ischaemia-reperfusion. The work was carried out at a high methodological level, and the reliability of the data is unquestionable. The data clearly show an important role for CCN2 in DNA damage processes and the development of oxidative stress in ischaemic tissues. The results are interesting and promising. However, I have a few comments.

Comments and Suggestions for Authors

  1. L.80. “Sample sizes were based on published studies and pilot experiments”. A reference is required after this sentence.
  2. L.106. “Cells were subjected to 24 hours of anoxia and 2 hours of reoxygenation”. Need to explain why these times are chosen, provide arguments + references.
  3. L.142. RNA-seq. “Full transcriptome RNA-sequencing was performed on a cDNA library constructed from RNA extracted from selected samples of kidney cortex”. You should indicate how many samples of renal tissue were taken for cDNA synthesis (for each group).
  4. Quantitative real-time PCR. Table S1. In the primers table, it would be helpful to list GenBank accession numbers and amplicon sizes for the corresponding genes.
  5. General remark on figures. I would like to see a more accurate presentation of the results. Pay attention to the axis labels. Columns can be colored in different colors to improve readability.
  6. Line 288 of the results contains the abbreviation CTGF, another name for CCN2. Earlier in the text, this abbreviation CTGF occurs only in lines 16 and 53. It is better to stick to the single notation CCN2.
  7. Fig.2. Where are the C, D panels?
  8. Fig.S2.B. The signatures on the ordinate axis have moved off. It's hard to make sense of it.
  9. Fig.3.B. The caption on the ordinate axis runs over the numbers.
  10. Fig.4. I would like to see a graph showing the % of cells in which the signal from Bcl-xL was detected, as was done for p-p53 and γH2AX.
  11. L.287. CCN2 deletion reduces DDR and apoptosis resistance 3 days after IRI. What was the survival rate of the animals by the 3rd day of the experiment (after 25 minutes of bilateral renal ischemia-reperfusion)?
  12. Figure 7. In my opinion, representing MDA concentration as nmol/mg protein is preferable.
  13. L.359. “CCN2 injected kidneys showed an increase in 4HNE expression compared to control mice.” From this proposal, it can be assumed that CCN2 was injected directly into the kidney rather than intraperitoneally.
  14. L.443. “Alternatively, reduced CCN2 might improve the handling of ROS by antioxidant  mechanisms. However, the fact that we observed reduced rather than increased expression of Nrf2 target genes HMOX1 and NQO1 in CCN2 depleted mice, indicates that the  reduced oxidative stress response may more likely reflect impaired generation of ROS rather than enhanced antioxidant mechanisms.”  In my opinion, it is difficult to talk about the effect of CCN2 knockout on the redox system by these two genes alone. Maybe the authors have ideas about the change in expression levels of such classic antioxidant response genes: SOD1-3, CAT, GPx, Prdx, etc. after CCN2 knockout.

Round 2

Reviewer 1 Report

All the issues have been addressed.